# De Bruijn goes Neural: Causality-Aware Graph Neural Networks for Time Series Data on Dynamic Graphs

**Lisi Qarkaxhija**
Chair of Machine Learning for Complex Networks
Center for Artificial Intelligence and Data Science (CAIDAS)
Julius-Maximilians-Universität Würzburg, DE
`lisi.qarkaxhija@uni-wuerzburg.de`

**Vincenzo Perri**
Data Analytics Group
Department of Informatics
University of Zurich, CH
`perri@ifi.uzh.ch`

**Ingo Scholtes**[*]
Chair of Machine Learning for Complex Networks
Center for Artificial Intelligence and Data Science (CAIDAS)
Julius-Maximilians-Universität Würzburg, DE
`ingo.scholtes@uni-wuerzburg.de`

## Abstract

We introduce De Bruijn Graph Neural Networks (DBGNNs), a novel time-aware graph neural network architecture for time-resolved data on dynamic graphs. Our approach accounts for temporal-topological patterns that unfold in the causal topology of dynamic graphs, which is determined by *causal walks*, i.e. temporally ordered sequences of links by which nodes can influence each other over time. Our architecture builds on multiple layers of higher-order De Bruijn graphs, an iterative line graph construction where nodes in a De Bruijn graph of order $k$ represent walks of length $k-1$, while edges represent walks of length $k$. We develop a graph neural network architecture that utilizes De Bruijn graphs to implement a message passing scheme that considers non-Markovian characteristics of causal walks, which enables us to learn patterns in the causal topology of dynamic graphs. Addressing the issue that De Bruijn graphs with different orders $k$ can be used to model the same data, we apply statistical model selection to determine the optimal graph to be used for message passing. An evaluation in synthetic and empirical data sets suggests that DBGNNs can leverage temporal patterns in dynamic graphs, which substantially improves performance in a node classification task.

## 1 Introduction

Graph Neural Networks (GNNs) [1, 2] are a cornerstone for applications of deep learning to data with a non-Euclidean, relational structure. Different flavors of GNNs have been shown to be highly efficient for tasks like node classification, representation learning, link prediction, cluster detection, or graph classification. The popularity of GNNs is largely due to the abundance of data that can be represented as graphs, i.e. as sets of *nodes* with pairwise connections represented as *links*. However, we increasingly have access to *time-stamped data* that not only capture which nodes are connected to each other, but also at which discrete points in time and in which temporal order those connections occur. A number of works in computer science, network science, and physics have highlighted how the *temporal dimension* of such *dynamic graphs* influences the *causal topology* of networked systems, i.e. which nodes can possibly influence each other over time [3–5]. In a nutshell, if an undirected link $(a, b)$ between two nodes $a$ and $b$ occurs *before* an undirected link $(b, c)$, node $a$ can causally influence node $c$ via node $b$. If the temporal ordering of those two links is reversed, node $a$ cannot

---

[*]also with Data Analytics Group, Department of Informatics, University of Zurich, Zurich, CH

Lisi Qarkaxhija et al., De Bruijn goes Neural: Causality-Aware Graph Neural Networks for Time Series Data on Dynamic Graphs. *Proceedings of the First Learning on Graphs Conference (LoG 2022)*, PMLR 198, Virtual Event, December 9–12, 2022.

influence node $c$ via $b$ due to the directionality of the arrow of time. This simple example shows that the arrow of time in dynamic graphs limits possible *causal* influences between nodes beyond what we expect based on the mere topology of links. In line with other uses of the term "causal" (e.g. in the context of *causal inference* [6] and graph learning [7]), the term "causal topology" is justified since the "correct" temporal ordering of links is a necessary (although not sufficient) condition for nodes to causally influence each other. Hence, the absence of a "causally" ordered sequence of interactions rules out the possibility of a causal influence between nodes, while the existence of such a sequence indicates the possibility (but not necessarily the presence) of causal influence.

Beyond such toy examples, a number of recent studies in network science, computer science, and interdisciplinary physics have shown that the temporal ordering of links in real time series data on graphs has non-trivial consequences for the properties of networked systems, e.g. reachability and percolation [8, 9], diffusion and epidemic spreading [10, 11], node rankings and community structures [12]. It had further been shown that this interesting aspect of dynamic graphs can be understood using a variant of *De Bruijn graphs* [13], i.e. static higher-order graphical models [11, 14, 15] of causal paths that capture both the temporal and the topological dimension of time series data on graphs.

While the generalization of network analysis techniques like node centrality measures and community detection [12, 14], or graph embedding [16] to such higher-order models has been successful, to the best of our knowledge no generalizations of Graph Neural Networks to higher-order De Bruijn graphs have been proposed [17, 18]. Such a generalization bears several promises: First it could enable us to apply well-known and efficient gradient-based learning techniques in a static neural network architecture that is able to learn patterns in the causal topology of dynamic graphs that are due to the temporal ordering of links. Second, making the temporal ordering of links in time-stamped data a first-class citizen of graph neural networks, this generalization could also be an interesting approach to incorporate a necessary condition for *causality* into state-of-the-art geometric deep learning techniques, which often lack meaningful ways to represent time. Finally, a combination of higher-order De Bruijn graph models with graph neural networks enable us to apply frequentist and Bayesian techniques to learn the "optimal" order of a De Bruijn graph model for a given time series, providing new ways to combine statistical learning and model selection with graph neural networks.

Addressing this gap, our work generalizes graph neural networks to high-dimensional De Bruijn graph models for causal paths in time-stamped data on dynamic graphs. We obtain a novel causality-aware graph neural network architecture for time series data that makes the following contributions:

- We develop a graph neural network architecture that generalizes message passing to multiple layers of higher-order De Bruijn graphs. The resulting De Bruijn Graph Neural Network (DBGNN) architecture is used to implement a message passing scheme, whose dynamics matches non-Markovian characteristics of causal walks, thus enabling us to learn patterns that shape the causal topology of dynamic graphs.
- We evaluate our DBGNN architecture in empirical and synthetic dynamic graphs and compare its performance to graph neural networks as well as (time-aware) graph representation learning techniques. We find that our method yields superior node classification performance.
- We combine this architecture with statistical model selection to infer the optimal higher order of a De Bruijn graph. This yields a two-step learning process, where (i) we first learn a parsimonious De Bruijn graph model that neither under- nor overfits patterns in a dynamic graph, and (ii) we apply message passing and gradient-based optimization to the inferred graph in order to address graph learning tasks like node classification or representation learning.

Our work builds on the –to the best of our knowledge– novel combination of (i) statistical model selection to infer optimal higher-order graphical models for dynamic graphs, and (ii) gradient-based learning in a GNN architecture that uses the inferred higher-order graphical models as message passing layers. Thanks to this approach, our architecture performs message passing in an optimal graph model for the causal paths in a given dynamic graph. The results of our evaluation confirm that this explicit regularization of the message passing layers enables us to considerably improve performance in a node classification task. The remainder of this paper is structured as follows: In section 2 we introduce the background of our work and formally state the problem that we address, in section 3 we introduce the De Bruijn graph neural network architecture, in section 4 we experimentally validate our method in synthetic and empirical data on dynamic graphs, and in section 5 we summarize our contributions and highlight opportunities for future research. We have

implemented our architecture based on the graph learning library `pyTorch Geometric` [19] and the higher-order graph analysis library `pathpy` [20]. We make our code available online [21].

## 2   Background and Problem Statement

**Basic definitions**   We consider a dynamic graph $G^{\mathcal{T}} = (V, E^{\mathcal{T}})$ with a (static) set of nodes $V$ and time-stamped (directed) edges $(v, w; t) \in E^{\mathcal{T}} \subseteq V \times V \times \mathbb{N}$ where –without loss of generality– integer timestamps $t$ represent the instantaneous time at which a pair of nodes $v, w$ is connected [4]. While many real-world network data exhibit such timestamps, for the application of graph neural networks we often consider a *time-aggregated projection* $G(V, E)$ along the time axis, where a (static) edge $(v, w) \in E$ exists iff $\exists t \in \mathbb{N} : (v, w; t) \in E^{\mathcal{T}}$. We can further consider edge weights $w : E \to \mathbb{N}$ defined as $w(v, w) := |\{t \in \mathbb{N} : (v, w; t) \in E^{\mathcal{T}}\}|$, i.e. we use $w(v, w)$ to count the number of temporal activations of $(v, w)$.

A key motivation for the study of graphs as models for complex systems is that –apart from *direct* interactions captured by edges $(v, w)$– they facilitate the study of *indirect* interactions between nodes via *paths* or *walks* in a graph. Formally, we define a walk $v_0, v_1, \ldots, v_{l-1}$ of length $l$ in a graph $G = (V, E)$ as any sequence of nodes $v_i \in V$ such that $(v_{i-1}, v_i) \in E$ for $i = 1, \ldots, l - 1$. The length $l$ of a walk captures the number of traversed edges, i.e. each node $v \in V$ is a walk of length zero, while each edge $(v, w)$ is a walk of length one. We further call a walk $v_0, v_1, \ldots, v_{l-1}$ a *path* of length $l$ from $v_0$ to $v_{l-1}$ iff $v_i \neq v_j$ for $i \neq j$, i.e. a path is a walk between a set of distinct nodes.

**Causal walks and paths in dynamic graphs**   In a static graph $G = (V, E)$, the topology–i.e. which nodes can *directly and indirectly* influence each other via edges, walks, or paths– is completely determined by the edges $E$. This is is different for dynamic graphs, which can be understood by extending the definition of walks and paths to *causal concepts* that respect the *arrow of time*:

**Definition 1.** *For a dynamic graph $G^{\mathcal{T}} = (V, E^{\mathcal{T}})$, we call a node sequence $v_0, v_1, \ldots, v_{l-1}$ a causal walk iff the following two conditions hold: (i) $\exists t_0, \ldots, t_{l-1} : (v_{i-1}, v_i; t_i) \in E^{\mathcal{T}}$ for $i = 1, \ldots, l - 1$ and (ii) $0 < t_j - t_i \leq \delta$ for $i < j$ and some $\delta > 0$.*

The first condition ensures that nodes in a dynamic graph can only indirectly influence each other via a causal walk iff a corresponding walk exists in the time-aggregated graph. Due to $0 < t_j - t_i$ for $i < j$, the second condition ensures that time-stamped edges in a causal walk occur in the correct chronological order, i.e. timestamps are monotonically increasing and thus respect the arrow of time [3, 4]. As an example, two time-stamped edges $(a, b; 1), (b, c; 2)$ constitute a causal walk by which information from node $a$ starting at time $t_1 = 1$ can reach node $c$ at time $t_2 = 2$ via node $b$, while the same edges in reverse temporal order $(a, b; 2), (b, c; 1)$ do not constitute a causal walk. While this definition of a causal walk does not impose an *upper bound* on the time difference between consecutive time-stamped edges constituting a causal walk, it is often reasonable to define a time limit $\delta > 0$, i.e. a time difference beyond which consecutive edges are not considered to contribute to a causal walk. As an example, two time-stamped edges $(a, b; 1), (b, c; 100)$ constitute a causal walk by which information from node $a$ starting at time $t_1 = 1$ can reach node $c$ at time $t_2 = 100$ via node $b$ for $\delta = 150$, while they do not constitute a causal walk for $\delta = 5$. This time-limited notion of causal or time-respecting walks is characteristic for many real networked systems in which processes or agents have a finite time scale or "memory", which rules out infinitely long gaps between consecutive causal interactions [4, 5]. Analogous to the definition in a static network, we finally define a *causal path* $v_0, v_1, \ldots, v_{l-1}$ of length $l$ from node $v_0$ to node $v_{l-1}$ as a causal walk with $v_i \neq v_j$ for $i \neq j$.

The definitions above have important consequences for our understanding of the *causal topology* of dynamic graphs, i.e. which nodes can possibly influence each other directly or indirectly via causal walks or paths. The causal topology of a static graph $G = (V, E)$ can be fully understood based on the *transitive hull of edges*, i.e. the presence of two edges $(u, v) \in E$ and $(v, w) \in E$ implies that nodes $u$ and $w$ can indirectly influence each other via a walk or path, which we denote as $u \to^* w$. This is the basis of graph analytic methods, e.g. to calculate (shortest) paths, eigenvalues and eigenvectors to analyze topological properties. In contrast, the chronological order of edges in dynamic graphs can break transitivity, i.e. $(u, v; t) \in E$ and $(v, w; t') \in E$ does *not* necessarily imply $u \to^* w$, which invalidates graph analytic approaches [15, 22].

To study how the temporal ordering of edges influences the *causal topology* of dynamic graphs, we can consider causal walks as sequences of random variables that can be modelled via Markov

chains of order $k$ over a discrete state space $V$ [14]. We thus model node sequences $v_0, \ldots, v_{l-1}$ in causal walks as $P(v_i|v_{i-k}, \ldots, v_{i-1})$, where $k-1$ is the "memory" of the model. For $k = 1$ we have a memoryless, first-order Markov chain $P(v_i|v_{i-1})$, where the next node on a walk only depends on the current node. This corresponds to causal walks that are determined by the topology (and possibly frequency) of edges, i.e. in absence of correlations in the temporal ordering of edges the causal topology of dynamic graphs corresponds to the time-aggregated graph. For order $k > 1$ the sequence of nodes traversed by causal walks exhibits *memory*, i.e. the next node on a walk can depend on past interactions. Such temporal correlations in dynamic graphs can result in complex causal topologies that (i) cannot be understood based on the time-aggregated graph, and (ii) influence spectral properties [11, 15, 23, 24], node centralities [14, 22, 25], and communities [12].

**Higher-order De Bruijn graph models of causal topologies** The use of higher-order Markov chain models for causal paths leads to a novel view, where the common (weighted) time-aggregated graph representation of time-stamped edges corresponds to a *first-order graphical model*, where edge weights capture the statistics of edges, i.e. causal paths of length one. A normalization of edge weights in this graph yields a first-order Markov model of causal walks in a dynamic graph. Similarly, a graphical representation of higher-order Markov chain model of causal walks can be used to capture non-Markovian patterns in the temporal sequence of time-stamped edges. However, different from higher-order Markov chain models of general categorical sequences, a higher-order model of causal paths in dynamic graphs must account for the fact that the set of possible *causal paths* is constrained by the topology of the corresponding static graph (i.e. condition (i) in Definition 1). To account for this we define a higher-order De Bruijn graph model of causal walks [13]:

**Definition 2** ($k$-th order De Bruijn graph). *For dynamic graph $G^{\mathcal{T}} = (V, E^{\mathcal{T}})$ and $k \in \mathbb{N}$, a $k$-th order De Bruijn graph model of causal paths in $G^{\mathcal{T}}$ is a graph $G^{(k)} = (V^{(k)}, E^{(k)})$, with $u := (u_0, u_1, \ldots, u_{k-1}) \in V^{(k)}$ a causal walk of length $k-1$ in $G^{\mathcal{T}}$ and $(u, v) \in E^{(k)}$ iff (i) $v = (v_1, \ldots, v_k)$ with $u_i = v_i$ for $i = 1, \ldots, k-1$ and (ii) $u \oplus v = (u_0, \ldots, u_{k-1}, v_k)$ a causal walk of length $k$ in $G^{\mathcal{T}}$.*

Any two adjacent nodes $u, v \in V^k$ in a $k$-th order De Bruijn graph $G^{(k)}$ represent two causal walks of length $k-1$ that overlap in exactly $k-1$ nodes, i.e. each edge $(u, v) \in E^{(k)}$ represents a causal walk of length $k$. We use edge weights $w : E^{(k)} \to \mathbb{N}$ to capture frequencies of causal paths of length $k$. The (weighted) time-aggregated graph $G$ of a dynamic graph trivially corresponds to a first-order De Bruijn graph, where (i) nodes are causal walks of length zero and (ii) edges $E = E^{(1)}$ capture causal walks of length one (i.e. edges) in $G^{\mathcal{T}}$. To construct a second-order De Bruijn graph $G^{(2)}$ we can perform a line graph transformation of a static graph $G = G^{(1)}$, where each edge $(u_0, u_1), (u_1, u_2) \in E^{(2)}$ captures a causally ordered sequence of two edges $(u_0, u_1; t)$ and $(u_1, u_2; t')$. A $k$-th order De Bruijn graph can be constructed by a repeated line graph transformation of a static graph $G$. Hence, De Bruijn graphs can be viewed as generalization of common graph models to a higher-order graphical model of causal walks of length $k$, where walks of length $l$ in $G^k$ model causal walks of length $k + l - 1$ in $G^{\mathcal{T}}$ [11, 15].

De Bruijn graphs have interesting mathematical properties that connect them to trajectories of subshifts of finite type and to dynamical systems and ergodic theory [26]. In our work, we use a $k$-th order De Bruijn graph to model the *causal topology* of dynamic graphs. We illustrate this in fig. 1, which shows two dynamic graphs with four nodes and 33 time-stamped links. These dynamic graphs only differ in the temporal ordering of edges, i.e. their (first-order) weighted graph representation is the same (center). Moreover, this first-order representation wrongly suggests that node $A$ can causally influence node $C$ by a path via node $B$. While this is true in the dynamic graph on the right (see red causal paths), no corresponding causal path from $A$ via $B$ to $C$ exists in the dynamic graph on the left. A second-order De Bruijn graph model (bottom left and right) captures the fact that the causal path from $A$ via $B$ to $C$ is absent in the right example. This shows that, different from commonly used static graph representations, the edges of a $k$-th order De Bruijn graph with $k > 1$ are sensitive to the temporal ordering of time-stamped edges. Hence, static higher-order De Bruijn graphs can be used to model the causal topology in a dynamic graph. We can view a $k$-th order De Bruijn graph in analogy to a $k$-th order Markov chain, where a directed link from state $(u_0, \ldots, u_{k-1})$ to state $(u_1, \ldots, u_k)$ captures a walk from node $u_{k-1}$ to $u_k$ in the underlying graph, with a memory of $k-1$ previously visited nodes $u_1, \ldots, u_{k-1}$. Similar higher-order graph models have been used to analyze how the causal topology of dynamic graphs influences node ranking [12, 14], random walks and diffusion [11], community detection [12, 23], as well as graph visualization and embedding [16, 27, 28].

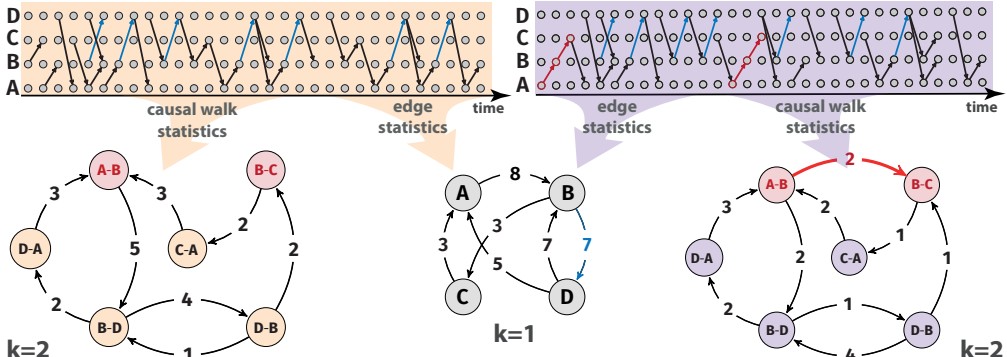

**Figure 1:** Example for two dynamic graphs with four nodes and 33 time-stamped edges (top left and right) that only differ in the temporal ordering of edges. Frequency and topology of edges are identical, i.e. they have the same first-order time-aggregated weighted graph (center). Due to the arrow of time, causal walks and paths differ in the dynamic graphs: Assuming $\delta = 1$, in the left example node $A$ cannot causally influence $C$ via $B$, while such a causal path is possible in the right example. A second-order De Bruijn graph model of causal walks in the two graphs (bottom left and right) captures this difference in the causal topology. Building on such higher-order graph models, we define a GNN architecture that is able to learn patterns in the causal topology of dynamic graphs.

**Problem Statement and Research Gap** The works above provide the background for the generalization of graph neural networks to higher-order De Bruijn graph models of causal walks in dynamic graphs, which we propose in the following section. Following the terminology in the network science community, higher-order De Bruijn graph models can be seen as one particular type of *higher-order network models* [15, 29, 30], which capture (causally-ordered) sequences of interactions between more than two nodes, rather than dyadic edges. They complement other types of popular higher-order network models (like, e.g. hypergraphs, simplicial complexes, or motif-based adjacency matrices) that consider (unordered) non-dyadic interactions in static networks, and which have been used to generalize graph neural networks to non-dyadic interactions [31, 32].

To the best of our knowledge, De Bruijn graph models have not been combined with recent advanced in graph neural networks. Closing this gap, we propose a causality-aware graph convolutional network architecture that uses an *augmented* message passing scheme [33] in higher-order De Bruijn graphs to capture patterns in the causal topology of dynamic graphs.

## 3   De Bruijn Graph Neural Network Architecture

We now introduce the De Bruijn Graph Neural Network (DBGNN) architecture with an *augmented message passing* [33] scheme whose dynamics matches the non-Markovian characteristics of causal walks in dynamic graphs, which is the key contribution of our work. While we build on the message passing proposed for Graph Convolutional Networks (GCN) [34], it is easy to generalize our architecture to other message passing schemes. Our approach is based on the following three steps, which yield an easy to implement and scalable class of graph neural networks for time series and sequential data on graphs: We first use time series data on dynamic graphs to calculate statistics of causal walks of different lengths $k$. We use these statistics to select an higher-order De Bruijn graph model for the causal topology of a dynamic graph. This step is parameter-free, i.e. we can use statistical learning techniques to infer an optimal graph model for the causal topology directly from time series data, without need for hyperparameter tuning or cross-validation. We now define a graph convolutional network that builds on neural message passing in the higher-order De Bruijn graphs inferred in step one. The hidden layers of the resulting graph convolutional network yield meaningful latent representations of patterns in the *causal topology of a dynamic graph*. Since the nodes in a $k$-th order De Bruijn graph model correspond to walks (i.e. sequences) of nodes of length $k-1$, we implement an additional bipartite layer that maps the latent space representations of sequences to nodes in the original graph. In the following, we provide a detailed description of these three steps.

**Inference of Optimal Higher-Order De Bruijn Graph Model**    The first step in the DBGNN architecture is the inference of the higher-order De Bruijn graph model for the causal topology in a given dynamic graph data set. For this, we use Definition 1 to calculate statistic of causal walks of different lengths $k$ for a given maximum time difference $\delta$. We note that this can be achieved using efficient window-based algorithms [35, 36]. The statistics of causal walks in the dynamic graph allows us to apply heuristic and statistical model selection techniques [12, 14, 25, 37] to find an optimal higher-order model given the statistics of causal walks (or paths). In our work, we employ the method proposed in [14], which yields the optimal higher order $k_{opt}$ for all data sets (see table 2). While for the details of the method we refer to [14], in the appendix we provide a high-level description of the approach. The resulting (static) higher-order De Bruijn graph model is the basis for our extension of the message passing scheme for dynamic graphs.

**Message passing in higher-order De Bruijn graphs**    Standard message passing algorithms in graph neural networks use the topology of a graph to propagate (and smooth) features across nodes, thus generating hidden features that incorporate patterns in the topology of a graph. To additionally incorporate patterns in the *causal topology of a dynamic graph* we perform message passing in multiple layers of higher-order De Bruijn graphs. Assuming a $k$-th order De Bruijn graph model $G^{(k)} = (V^{(k)}, E^{(k)})$ as defined in Definition 2, the input to the first layer $l = 0$ is a set of $k$-th order node features $\mathbf{h^{k,0}} = \{\vec{h}_1^{k,0}, \vec{h}_2^{k,0}, \ldots, \vec{h}_N^{k,0}\}$, for $\vec{h}_i^{k,0} \in \mathbb{R}^{H^0}$, where $N = |V^{(k)}|$ and $H^0$ is the dimensionality of initial node features. The De Bruijn graph message passing layer uses the causal topology to learn a new set of hidden representations for higher-nodes $\mathbf{h^{k,1}} = \{\vec{h}_1^{k,1}, \vec{h}_2^{k,1}, \ldots, \vec{h}_N^{k,1}\}$, with $\vec{h}_i^{k,1} \in \mathbb{R}^{H^1}$ for each $k - th$ order node $i$ (corresponding to a causal walk of length $k - 1$). For layer $l$, we define the update rule of the message passing as:

$$\vec{h}_v^{k,l} = \sigma\left(\boldsymbol{W}^{k,l} \sum_{\{u \in V^{(k)} : (u,v) \in E^{(k)}\} \cup \{v\}} \frac{w(u,v) \cdot \vec{h}_u^{k,l-1}}{\sqrt{S(v) \cdot S(u)}}\right), \tag{1}$$

where $\vec{h}_u^{k,l-1}$ is the previous hidden representation of node $u \in V^k$, $w(u,v)$ is the weight of edge $(u,v) \in E^k$ (capturing the frequency of the corresponding causal walk as explained in section 2), $\boldsymbol{W}^{k,l} \in \mathbb{R}^{H^l \times H^{l-1}}$ are trainable weight matrices, $S(v) := \sum_{u \in V^{(k)}} w(u,w)$ is the sum of weights of incoming edges of nodes, and $\sigma$ is a non-linear activation function. Since it is performed on a higher-order De Bruijn graph, we obtain a message passing that is influenced by the non-Markovian characteristics of causal walks in the underlying dynamic graph. Different from standard, static graph neural networks that ignore the temporal dimension of dynamic graphs, this enables our architecture to incorporate temporal patters that shape the causal topology, i.e. which nodes in a dynamic graph can influence each other directly and indirectly based on the temporal ordering of edges.

**First-order message passing and bipartite projection layer**    The (static) topology of a graph influences which causal walks are theoretically possible (i.e. they constitute valid walks in the graph) and thus which edges can exist in the $k$-th order De Bruin graph. However, since it operates on nodes $V^{(k)}$ in the *higher-order* graph, the message passing outlined above does not allow us to incorporate information on the first-order topology. To address this issue, we additionally include message passing in the (static) time-aggregated weighted graph $G$, which can be done in parallel to the message passing in the higher-order De Bruijn graph. The $g$ layers of this first-order message passing (whose formal definition we omit as it simply uses the GCN update rule [34]) generate hidden representations $\vec{h}_v^{1,g}$ of nodes $v \in V$. This approach enables us to incorporate optional node features $\vec{h}_v^{0,g}$ (or alternatively use a one-hot-encoding of nodes). In the appendix, we include an ablation study that highlights the advantage of first-order message passing.

Since the message passing in a higher-order De Bruijn graph generates hidden features for higher-order nodes $V^{(k)}$ (i.e. sequences of $k$ nodes) rather than nodes $V$ in the original dynamic graph, we finally define a bipartite graph $G^b = (V^{(k)} \cup V, E^b \subseteq V^{(k)} \times V)$ that maps node features of higher-nodes to the first-order node space. For a given node $v \in V$, this bipartite layer sums the hidden representations $\vec{h}_u^{k,l}$ of each higher-order node $u = (u_0, \ldots, u_{k-1}) \in V^{(k)}$ with $u_{k-1} = v$ to the representation $h_v^{1,g} \in \mathbb{R}^{F^g}$ generated by the last layer of the first-order message passing. The choice of a bipartite mapping that aggregates higher-order nodes $u$ based on the *last* first-order node $u_{k-1}$ is based on the interpretation of multiple $k$-th order De Bruijn graph message passing layers in

analogy to higher-order Markov chains, where subsequent layers "shift" the memory prefix by one position, while the last node captures the current state of the chain.

Notice that the dimensions of representations in the last layers of the $k$-th and first-order message passing should satisfy $F^g = H^l$ to enable the summing of the representations. We obtain representations $\{\vec{h}_u^{k,l} + \vec{h}_v^{1,g} : \text{for } u \in V^k \text{ with } (u,v) \in E^b\}$ that are the higher-order node representations augmented by the corresponding first order representations. We then use a function $\mathcal{F}$ to aggregate the augmented higher-order representations at the level of first-order nodes. In our experiments, we learn first-order node representations $h^{1,g}$ using GCN message passing with $g$ layers, allowing to integrate information on the static and the causal topology of a dynamic graph. Formally, we define the bipartite layer as

$$\vec{h}_v^b = \sigma \left( \boldsymbol{W}^b \mathcal{F} \left( \{\vec{h}_u^{k,l} + \vec{h}_v^{1,g} : \text{for } u \in V^{(k)} \text{ with } (u,v) \in E^b\} \right) \right), \qquad (2)$$

where $\vec{h}_v^b$ is the output of the bipartite layer for node $v \in V$, and $\mathbf{W}^b \in \mathbb{R}^{F^g \times H^l}$ is a learnable weight matrix. The function $\mathcal{F}$ can be SUM, MEAN, MAX, MIN.

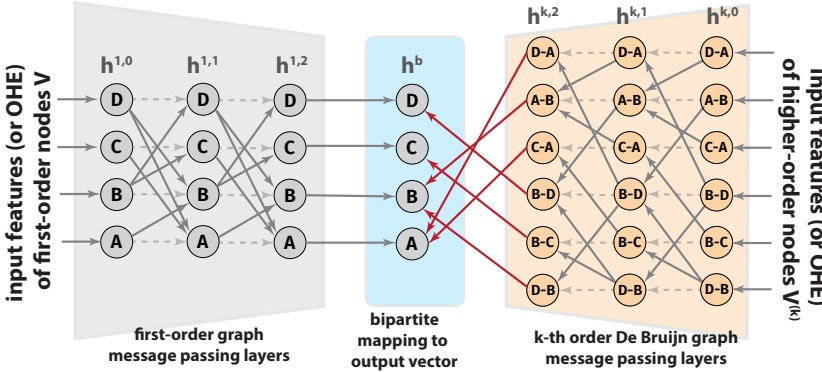

**Figure 2:** Illustration of DBGNN architecture with two message passing layers in first- (left, gray) and second-order De Bruijn graph (right, orange) corresponding to the dynamic graph in Figure 1 (left). Red edges represent indicate the bipartite mapping $G^b$ of higher-order node representations to first-order representations. An additional linear layer (not shown) is used for node classification.

Figure 2 gives an overview of the proposed neural network architecture for the dynamic graph (and associated second-order De Bruijn graph model) shown in Figure 1 (left). The higher-order message passing layers on the right use the topology of the second-order De Bruijn graph in Figure 1 (left), while the first-order message passing layers (left) use the topology of the first-order graph. Note that the first-order and higher-order message passing can be performed in parallel, and that the number of message passing layers do not necessarily need to be the same. Red edges indicate the propagation of higher-order node representations to first order nodes performed in the final bipartite layer. Due to space constraints, in Figure 2 we omit the final linear layer used for classification.

## 4   Experimental Evaluation

In the following, we experimentally evaluate our proposed causality-aware graph neural network architecture both in synthetic and empirical time series data on dynamic graphs. With our evaluation, we want to answer the following questions:

**Q1** How does the performance of De Bruijn Graph Neural Networks compare to temporal and non-temporal graph learning techniqes?

**Q2** Can we use De Bruijn Graph Neural Networks to learn interpretable *static* latent space representations of nodes in *dynamic* graphs?

To address those questions, we use six time series data sets on dynamic graphs that provide meta-information on node classes. The overall statistics of the data sets can be found in table 2, **temp-clusters** is a synthetically generated dynamic graph with three clusters in the *causal topology*, but

no pattern in the *static topology*. To generate this data set, we first constructed a random graph and generated random sequences of time-stamped edges. We then selectively swap the time stamps of edges such that causal walks of length two within three clusters of nodes are overrepresented, while causal walks between clusters are underrepresented. We include a more detailed description in the appendix (code and data will be provided in a companion repository).

Apart from this synthetic data set, we use five empirical time series data sets: **student-sms** captures time-stamped SMS exchanged over four weeks between freshmen at the Technical University of Denmark [38]. We use the gender of participants as ground truth classes and use a maximum time difference of $\delta = 40$. Since the time granularity of this data set is five minutes, this corresponds to a maximum time difference of 200 minutes. **high-school-2011** and **high-school-2012** capture time-stamped proximities between high-school students in two consecutive years [39] (4 days in 2001, 7 days in 2012). We use the gender of students as ground truth classes. **workplace** captures time-stamped proximity interaction between employees recorded in an office building for multiple days in different years [40]. We use the department of employees as ground truth classes. **hospital** captures time-stamped proximities between patients and healthcare workers in a hospital ward. We use employees' roles (patient, nurse, administrative, doctor) as ground truth node classes. All proximity data sets have a temporal resolution of 20 seconds.

To mitigate the computational complexity of the causal walk extraction in the (undirected) proximity data sets, we coarsen the resolution by aggregating interactions to a resolution of fifteen minutes and use $\delta = 4$, which corresponds to a maximum time difference of one hour. Based on the resulting statistics of causal walks, we use the method (and code) provided in [14] to select an optimal higher-order De Bruijn graph model. In Table 2 we report the optimal order $k_{opt}$ detected by likelihood ratio test, which is used to test the hypothesis that a first-order graph model is sufficient to explain the observed causal walk statistics, against the alternative hypothesis that a higher-order De Bruijn graph model is needed (see description of order detection in appendix).

One goal of the synthetic model described above is to show how a *static* latent property of nodes (here: cluster labels) can introduce patterns in the causal topology of a temporal network. Here, following an edge $(u, v)$ where node $u$ is member of a given community, node $v$ is more likely to next interact with another node in the same community, which influences the statistics of causal walks connecting nodes across communities. Similar mechanisms are likely at work in real temporal networks, e.g. node properties like gender, roles, or social groups can introduce correlations in the ordering of time-stamped interactions, which are expressed by the presence of second-order correlations in all five empirical data sets. The goal of our analysis is to show that such patterns can be used to improve the performance of node classification. To this end, we compare the node classification performance of the DBGNN architecture with an higher-order De Bruijn graph with optimal order $k_{opt}$ against the following five baselines. The first three are standard (static) graph learning techniques, namely Graph Convolutional Networks (**GCN**) [34], **DeepWalk** [41] and **node2vec** [42]. We further use two recently proposed temporal graph embedding techniques that capture non-Markovian dependencies in time series data on graphs. Embedding Variable Orders (**EVO**) [16] uses random walks to obtain vector representation of higher-order nodes. Then, a time-aware representation of each node is obtained by aggregating the vector representations of all the higher-order nodes that contain it. In HONEM [27], the causal paths observed in time-series data are used to populate a higher-order neighborhood matrix. Then, a time-aware node representations is obtained by applying truncated SVD to the higher-order neighborhood matrix. Due to the resemblance of a $k$-th order De Bruijn graph with the line graph of the $(k-1)$-th order graph (which can be viewed as "null model" capturing all possible walks of length $k$), we finally included **LGNN** [43], a generalization of GCN to line graphs, which we adopt to address node classification.

Addressing Q1, the results of our experiments on node classification are shown in Table 1. Since the classes of the empirical data sets are imbalanced, we use balanced accuracy and additionally report macro-averaged precision, recall and f1-score for a 70-30 training-test split.

The macro average performs the arithmetic mean of the scores (precision, recall and f1-score) obtained on the individual classes. We report the average performance across multiple splits. For DBGNN, GCN, LGNN, DeepWalk, node2vec, and HONEM we performed 50 runs. Due to its larger computational complexity (and time constraints) we could only perform 10 runs on EVO. The standard deviations are included in the appendix. We trained node2vec, EVO, and DeepWalk with 80 walks of length 40 per each node and a window of 10. We obtained the embeddings using the

word2vec implementation in [44]. For EVO, we use the average as an aggregator for the higher-order representations. To ensure the comparability of the results from GCN and DBGNN, we train both with the same number of convolutional layers with a learning rate of 0.001 for 5000 epochs, ELU [45] as activation function, and Adam [46] optimiser. For DBGNN, we use SUM as aggregation function $\mathcal{F}$. We used one-hot encoding of nodes as feature matrix (and one-hot encoding of higher-order nodes in the initial layer of the DBGNN). For all methods, we fix the dimensions of the last hidden layer to $d = 16$. We manually tuned the hidden dimensions of the first hidden layers for GCN and DBGNN, as well as the p and q parameters of EVO and node2vec. For LGNN we adjust the experimental setup in [47] for node classification task. We report results for the best combination of hyperparameters.

The results in Table 1 for the synthetic temporal clusters data set show that the three time-aware methods (EVO, HONEM, and DBGNN) perform considerably better than the static counterparts, which only "see" a random graph that does not allow to meaningfully assign node classes. Both EVO and our proposed DBGNN architecture are able to perfectly classify nodes in this data set. Despite their good performance in the synthetic data set, the three time-aware methods show much higher variability in the empirical data. DBGNN shows superior performance in terms of balanced accuracy, f1-macro, and recall-macro, for all empirical data sets, with relative performance increases compared to the second best method ranging from $1.55\%$ to $22.65\%$. For precision-macro, DBGNN performs best in four of the five. We attribute this to the ability of DBGNN to consider both patterns in the (static) graph topology and the causal topology, as well as to the underlying supervised approach.

| dataset | method | Balanced Accuracy | F1-score-macro | Precision-macro | Recall-macro |
|---|---|---|---|---|---|
| temp-clusters | DeepWalk | 32.47 | 30.39 | 32.25 | 32.47 |
| | Node2Vec p=1 q=4 | 35.48 | 33.02 | 34.92 | 35.48 |
| | GCN (8,32) | 33.52 | 12.5 | 8.61 | 33.52 |
| | EVO p=1 q=1 | **100.0** | **100.0** | **100.0** | **100.0** |
| | HONEM | 54.94 | 53.5 | 58.16 | 54.94 |
| | LGNN | 33.33 | 16.67 | 11.11 | 33.33 |
| | DBGNN (16,16) | **100.0** | **100.0** | **100.0** | **100.0** |
| gain | | 0% | 0% | 0% | 0% |
| high-school-2011 | DeepWalk | 55.25 | 54.02 | 60.45 | 55.25 |
| | Node2Vec p=1 q=4 | 56.89 | 56.29 | 60.05 | 56.89 |
| | GCN (32,4) | 50.06 | 40.27 | 33.99 | 50.06 |
| | EVO p=1 q=4 | 57.21 | 56.28 | 62.09 | 57.21 |
| | HONEM | 54.24 | 53.08 | 56.44 | 54.24 |
| | LGNN | 52.76 | 46.16 | 51.10 | 52.76 |
| | DBGNN (32,8) | **64.4** | **63.7** | **65.14** | **64.4** |
| gain | | 12.57% | 13.16% | 4.91% | 12.57% |
| high-school-2012 | DeepWalk | 59.46 | 59.6 | 71.71 | 59.46 |
| | Node2Vec p=1 q=4 | 60.75 | 61.23 | **72.44** | 60.75 |
| | GCN (8,32) | 58.03 | 56.39 | 59.16 | 58.03 |
| | EVO p=4 q=1 | 57.98 | 57.5 | 69.42 | 57.98 |
| | HONEM | 53.16 | 51.7 | 56.59 | 53.16 |
| | LGNN | 50.56 | 41.27 | 46.26 | 50.56 |
| | DBGNN (4,8) | **65.8** | **65.89** | 67.27 | **65.8** |
| gain | | 8.31% | 7.61% | -7.14% | 8.31% |
| hospital | DeepWalk | 47.18 | 44.18 | 43.91 | 47.18 |
| | Node2Vec p=1 q=4 | 50.6 | 47.14 | 45.81 | 50.6 |
| | GCN [32,32] | 49.48 | 44.62 | 43.55 | 49.48 |
| | EVO p=1 q=4 | 36.34 | 36.44 | 42.1 | 36.34 |
| | HONEM | 46.17 | 43.13 | 44.45 | 46.17 |
| | LGNN | 30.90 | 21.30 | 17.52 | 30.91 |
| | DBGNN (32,16) | **59.73** | **55.81** | **56.19** | **59.07** |
| gain | | 18.04% | 18.29% | 22.65% | 16.64% |
| student-sms | DeepWalk | 53.22 | 50.57 | 60.57 | 53.22 |
| | Node2Vec p=1 q=4 | 53.22 | 50.97 | 58.56 | 53.22 |
| | GCN (4,32) | 57.33 | 57.25 | 57.72 | 57.33 |
| | EVO p=4 q=1 | 52.93 | 50.66 | 57.14 | 52.93 |
| | HONEM | 50.43 | 44.44 | 52.91 | 50.43 |
| | LGNN | 52.40 | 52.11 | 52.33 | 52.39 |
| | DBGNN (4,4) | **60.6** | **60.89** | **62.55** | **60.6** |
| gain | | 5.7% | 6.36% | 3.27% | 5.7% |
| workplace | DeepWalk | 77.81 | 76.74 | 76.06 | 77.81 |
| | Node2Vec p=1 q=4 | 78.0 | 77.01 | 76.38 | 78.0 |
| | GCN (32,16) | 81.86 | 78.72 | 78.58 | 79.93 |
| | EVO p=1 q=4 | 77.0 | 75.68 | 75.03 | 77.0 |
| | HONEM | 73.26 | 72.82 | 73.73 | 73.26 |
| | LGNN | 28.38 | 17.90 | 16.64 | 28.38 |
| | DBGNN (32,8) | **83.13** | **81.06** | **81.52** | **81.75** |
| gain | | 1.55% | 2.97% | 3.74% | 2.28% |

**Table 1:** Results of node classification for static graph learning techniques (DeepWalk, node2vec, GCN, LGNN), time-aware methods (HONEM, EVO) and the proposed DBGNN architecture.

To address Q2, we study hidden representations of higher- and first-order nodes generated by the DBGNN architecture for the synthetic data set, which exhibits three clusters in the causal topology. We use the hidden representations $\vec{h_v^b}$ generated by the bipartite layer of our DBGNN architecture, as defined in Section 3. We compare this to the representation generated by GCN. Figure 3 confirms that the DBGNN architecture learns meaningful node representations that incorporate temporal patterns.

## 5   Conclusion

We propose a new way to apply GNNs to data that capture the temporal ordering of edges in dynamic graphs. Our method is based on a combination of (i) a statistical approach to infer an optimal static higher-order De Bruijn graph model for the causal topology that is due to the temporal ordering of edges, (ii) gradient-based learning in a neural network architecture that performs neural message passing in the inferred higher-order De Bruijn graph, and (iii) a bipartite mapping layer that maps the learnt hidden representation of higher-order nodes to the original node space. Thanks to this approach, our architecture is able to generalize neural message passing to a *static* higher-order graph model that captures the causal topology of a dynamic graph, which can considerably deviate from what we would expect based on the mere (static) topology of edges. The results of our experiments demonstrate that the resulting architecture considerably improves the performance of node classification in time series data, despite using message passing in a simple static (augmented) graph.

Our work provides potential for several follow-up studies: First, our method requires to select a parameter $\delta$, which determines the "time scale" of the patterns considered by our model. Automatically learning the most "informative" time scale for a given temporal network is an interesting and open problem that we currently address in a separate work. Second, while for the present work we have focused on node classification, it is reasonable to assume that the DBGNN architecture can be used for other graph learning tasks like link prediction or graph classification. The clusters recovered in the learned latent space representations for the synthetic data further suggest a possible application to community detection in temporal networks. Finally, the DBGNN architecture allows to include node or edge features to further improve node classification performance. Since our work focuses on the advantage of considering the temporal ordering of edges, here we did not consider data with additional node or features, which can be addressed in future work.

Bridging recent research on higher-order graph models in network science and deep learning in graphs [15, 17, 29, 30], our work contributes to the ongoing discussion about the need for *augmented message passing* schemes in data on graphs with complex characteristics [33]. Moreover, we use statistical learning techniques to learn an optimal higher-order graph model for the causal topology of a dynamic graph that is then used to apply message passing in a graph neural network architecture. Referring to recent discussions in the network science community [48], this approach naturally distinguishes between (i) which data have been observed for a networked system, and (ii) what is the best model to parsimoniously represent those observations.

**Acknowledgements**   Vincenzo Perri and Ingo Scholtes acknowledge support by the Swiss National Science Foundation, grant 176938. Lisi Qarkaxhija thanks Chester Tan for useful discussions.

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

## A   Overview of data sets

| Data set | Ref | $|V|$ | $|E|$ | $|E^{\mathcal{T}}|$ | $k_{opt}$ | $|V^{(2)}|$ | $|E^{(2)}|$ | $\delta$ | Classes |
|---|---|---|---|---|---|---|---|---|---|
| temp-clusters | [49] | 30 | 560 | 60000 | 2 | 560 | 6,789 | 1 | 3 |
| high-school-2011 | [39] | 126 | 3042 | 28561 | 2 | 3042 | 17141 | 4 | 2 |
| high-school-2012 | [39] | 180 | 3965 | 45047 | 2 | 3965 | 20614 | 4 | 2 |
| hospital | [50] | 75 | 2028 | 32424 | 3 | 2028 | 15500 | 4 | 4 |
| student-sms | [38] | 429 | 733 | 46138 | 2 | 733 | 846 | 40 | 2 |
| workplace | [40] | 92 | 1431 | 9827 | 2 | 1431 | 7121 | 4 | 5 |

**Table 2:** Overview of time series data and ground truth node classes used in the experiments.

## B   Description of order detection technique

In our work, we use the order detection technique that was proposed and evaluated in [14]. The method casts the selection of the optimal order as a statistical model selection problem that can be addressed using a likelihood ratio test. For a given temporal network, it uses the statistics of observed causal walks to calculate the likelihood of a probabilistic De Bruijn graph model with maximum order $k$. A likelihood ratio test is then used to test the goodness of a fit of a $k$-th order model compared to a more complex model with order $k + 1$, while accounting for the difference in model complexity between the $k$ and $(k + 1)$-th order De Bruijn graph. The null hypothesis that a $k$-th order model is sufficient to explain the statistics of observed causal walk is rejected in favor of the alternative hypothesis that a $(k + 1)$-th model is needed if the associated increase in model likelihood is larger than the sampling error, which accounts both for the difference in model complexity and the amount of available data. This test can be efficiently be performed using Wilks' theorem for nested models [14]. With this, an incremental test of a $k$-th order (null model) against a model with order $k + 1$ allows us to select the a De Bruijn graph with the largest order $k$ that is justified given the complexity of the model and the available data. Specifically, it can be used to select any optimal order $k_{opt} \geq 1$, i.e. we can not only distinguish between first- and second-order De Bruijn graphs.

## C   Generation of Synthetic data with temporal clusters

**temp-clusters** is a synthetically generated dynamic graph with a random static topology but a strong cluster structure in the causal topology. To generate the dynamic graph, we first generate a static directed random graph with $n$ vertices and $m$ edges. For our experiment we chose $n = 30$ and $m = 560$. We randomly assign vertices to three equally-sized, non-overlapping clusters, where $C(v)$ denotes the cluster of vertex $v$. We then generate $N$ sequences of two randomly chosen time-stamped edges $(v_0, v_1; t)$ and $(v_1, v_2; t + 1)$ that contribute to a causal walk of length two in the resulting dynamic graph. For each vertex $v_1$ of such a causal path of length two, we randomly pick:

- two time-stamped edges $(u, v_1; t_1)$ and $(v_1, w, t_1 + 1)$ such that $C(u) = C(v_1) \neq C(w)$
- two time-stamped edges $(x, v_1; t_2)$ and $(v_1, z; t_2 + 1)$ with $C(v_1) = C(z) \neq C(x)$

Finally, we swap the time stamps of the four time-stamped edges to $(u, v_1; t_1)$ and $(v_1, z; t_1 + 1)$, $(x, v_1, t_2)$, and $(v_1, w, t_2 + 1)$. This swapping procedure is repeated for each vertex $v_1$ of a causal path of length two. This simple process changes the temporal ordering of time-stamped edges, affecting neither the topology nor the frequency of time-stamped edges. The model changes time stamps of edges (and thus causal paths) such that vertices are preferentially connected—via causal paths of length two—to other vertices in the same cluster. This leads to a strong cluster structure in the causal topology of the dynamic graph, which (i) is neither present in the time-aggregated topology nor in the temporal activation patterns of edges, and (ii) can nevertheless be detected by higher-order methods. A random reshuffling of timestamps destroys the cluster pattern, which confirms that it is only due to the temporal order of time-stamped edges.

The larger gain of our DBGNN architecture for the synthetically generated data compared to the empirical data sets (observed in Table 1) is likely due to the fact that the synthetic model purposefully generates strong patterns that allow to accurately classify nodes. This demonstrates the type of patterns that can be used by our architecture, while real data likely exhibit a mix of patterns that influence node classification.

## D    Latent Space Embeddings of Synthetic Example

Figure 3 shows a latent representation of nodes in the synthetic data set temp-clusters generated by the DBGNN (a) and GCN (b) architecture. This synthetically generated dynamic graph contains no pattern whatsoever in the (static) graph topology, which corresponds to a random graph, i.e. the topology of edges is random and all nodes have similar degrees (cf. Figure 3(b)). However, correlations in the temporal ordering of edges lead to three strong clusters in the *causal topology*, i.e. there are three groups of nodes where –due to the arrow of time and the temporal ordering of edges– pairs of nodes within the same cluster can influence each other via causal walks more frequently than pairs of nodes in different clusters. We emphasize that the resulting pattern in the causal topology is exclusively due to the temporal ordering of edges. The latent space embedding in Figure 3(a) highlights the DBGNN architectures's ability to learn this pattern in the causal topology of the underlying dynamic graph, which is absent in Figure 3(b). As expected, the different node degrees of the static graph (visible as clusters in Figure 3(b)) are the only pattern captured in the hidden node representations of the GCN architecture, which is insensitive to the temporal ordering of edges. This synthetic example confirms that DBGNNs provide a simple, static causality-aware approach for deep learning in dynamic graphs.

The fact that the learned latent representation of nodes clearly and accurately capture the known community structure encoded in the causal topology of the synthetic temporal network indicates that our method could be used for community detection, a potential that we seek to explore in future works.

## E    Semi-supervised Node Classification in DBGNNs

As an additional experiment that highlights the potential of our method beyond supervised node classification, we have performed an experiment where we apply the DBGNN model to the temporal clusters data set to address a *semi-supervised node classification task*.

For this illustrative experiment, we have chosen one random node per community in the training set (see grey nodes in Figure 4) and applied (i) the causality-aware DBGNN architecture and (ii) the GCN architecture to classify the remaining nodes in a test set. The latent representations of test nodes, where node colors capture the predicted class labels, are shown for the DBGNN and the GCN architecture in Figure 4(a) and Figure 3(b) respectively. We find that all predicted labels are correct for the DBGNN architecture, which is also visible based on the correlation between latent space positions of nodes and node predictions (see Figure 4(a)). In contrast, the predictions of the GCN architecture are not better than a random guess (see Figure 3(b)). To facilitate the comparison of predicted node labels, we have used the DBGNN latent space representations of nodes for both Figure 4(b) and Figure 4(a).

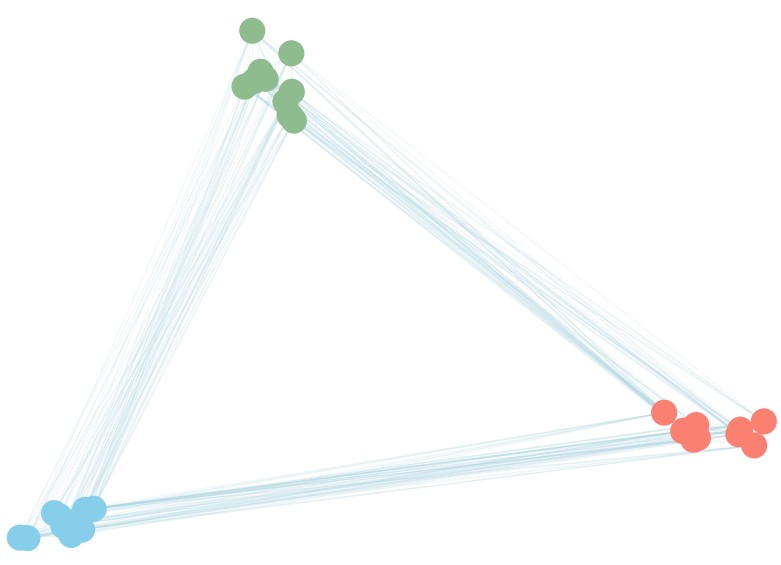

(a) Latent space representation of nodes generated by De Bruijn Graph Neural Network (DBGNN) using higher-order De Bruijn graph with order $k = 2$.

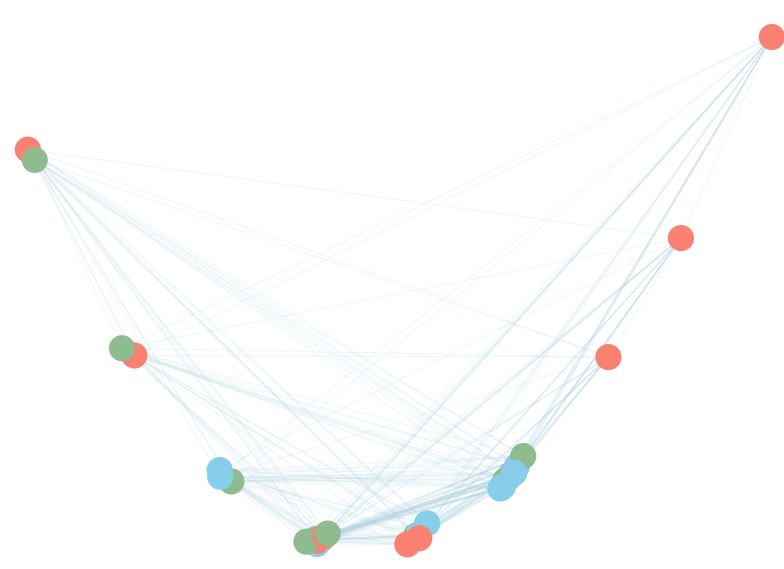

(b) Latent space representation of nodes generated by Graph Convolutional Network (GCN).

**Figure 3:** Latent space representations of nodes in a synthetically generated dynamic graph (**temp-clusters**) with three clusters in the causal topology, where colours indicate cluster memberships. The hidden node representations learned by the DBGNN architecture capture the cluster structure in the causal topology, which is exclusively due to the temporal ordering –and not due to the topology or frequency– of time-stamped edges.

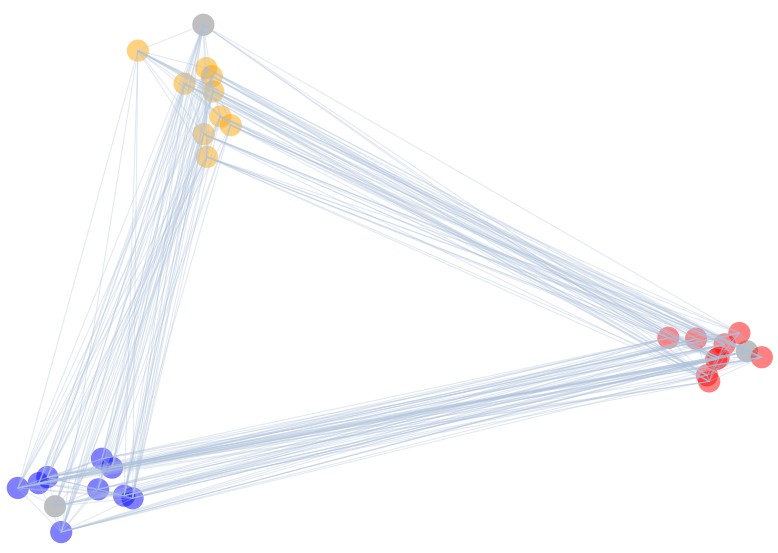

(a) Semi-supervised node classification using DBGNN (where nodes are positioned based on DBGNN latent representation)

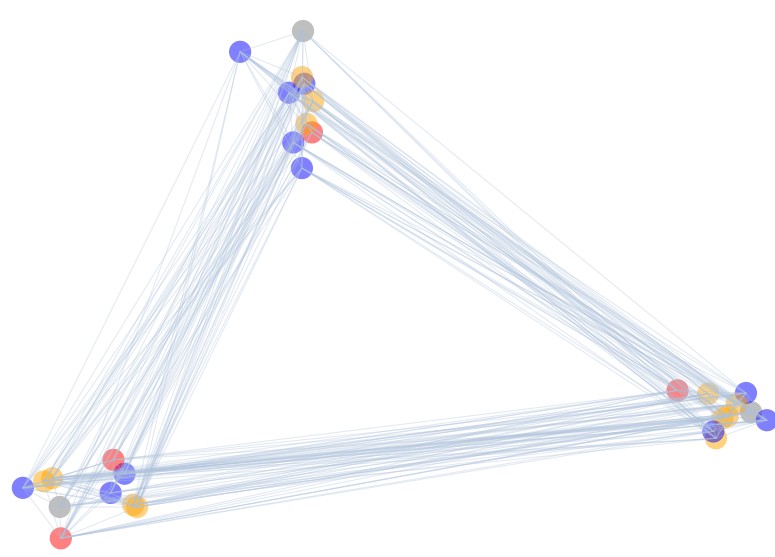

(b) Semi-supervised node classification using GCN (where nodes are positioned based on DBGNN latent representation)

**Figure 4:** Semi-supervised node classification with DBGNN (a) and GCN (b) for the synthetically generated dynamic graph (**temp-clusters**). Node colors indicate predicted node classes, where training nodes are shown in gray. For clarity of exposition, in both panels (a) and (b) we used the PCA reduced hidden representation from DBGNN for the node layout. In both panels we highlight the seed nodes in grey. The mixture of node colors in (b) demonstrates GCN inability to recover the ground truth node classes. As highlighted by the agreement in colors and node clusters in (a), DBGNN perfectly recovers the ground truth classes.

## F   Standard Deviation of Classification Results

In Table 3 we present the standard deviation of the classification results reported in table 1 across all runs for all models.

| dataset | method | Balanced Accuracy | F1-score-macro | Precision-macro | Recall-macro |
|---|---|---|---|---|---|
| temp-clusters | DeepWalk | 15.38 | 15.04 | 18.03 | 15.38 |
| | Node2Vec p=1 q=4 | 17.12 | 16.88 | 20.24 | 17.12 |
| | GCN (8,32) | 7.3 | 7.69 | 8.04 | 7.3 |
| | EVO p=1 q=1 | 0.0 | 0.0 | 0.0 | 0.0 |
| | HONEM | 16.27 | 16.71 | 19.61 | 16.27 |
| | LGNN | 0 | 0 | 0 | 0 |
| | DBGNN (16,16) | 0.0 | 0.0 | 0.0 | 0.0 |
| high-school-2011 | DeepWalk | 5.83 | 7.22 | 12.79 | 5.83 |
| | Node2Vec1.04.0 | 6.34 | 7.58 | 9.44 | 6.34 |
| | GCN (32,4) | 0.89 | 3.1 | 4.83 | 0.89 |
| | EVO p=1 q=4 | 5.72 | 7.65 | 9.33 | 5.72 |
| | HONEM | 5.72 | 6.93 | 10.07 | 5.72 |
| | LGNN | 6.41 | 10.48 | 17.22 | 6.42 |
| | DBGNN (32,8) | 7.0 | 7.42 | 7.8 | 7.0 |
| high-school-2012 | DeepWalk | 4.97 | 6.52 | 11.0 | 4.97 |
| | Node2Vec p=1 q=4 | 5.27 | 6.8 | 11.29 | 5.27 |
| | GCN (8,32) | 6.87 | 9.49 | 13.58 | 6.87 |
| | EVO p=4 q=1 | 4.14 | 6.07 | 9.96 | 4.14 |
| | HONEM | 4.59 | 5.89 | 9.12 | 4.59 |
| | LGNN | 4.48 | 10.75 | 14.91 | 4.48 |
| | DBGNN (4,8) | 6.59 | 6.62 | 7.07 | 6.59 |
| hospital | DeepWalk | 7.64 | 6.9 | 7.51 | 7.64 |
| | Node2Vec p=1 q=4 | 6.79 | 6.46 | 6.95 | 6.79 |
| | GCN (32,32) | 11.06 | 12.0 | 13.58 | 11.06 |
| | EVO p=1 q=4 | 9.31 | 11.34 | 16.31 | 9.31 |
| | HONEM | 8.51 | 7.78 | 8.25 | 8.51 |
| | LGNN | 7.38 | 10.65 | 10.76 | 7.38 |
| | DBGNN (32,16) | 10.7 | 9.68 | 9.9 | 10.74 |
| student-sms | DeepWalk | 2.72 | 4.45 | 10.05 | 2.72 |
| | Node2Vec p=1 q=4 | 3.29 | 4.93 | 9.13 | 3.29 |
| | GCN (4,32) | 3.59 | 3.65 | 3.91 | 3.59 |
| | EVO p=4 q=1 | 3.38 | 5.14 | 7.89 | 3.38 |
| | HONEM | 1.29 | 2.31 | 15.0 | 1.29 |
| | LGNN | 4.46 | 4.44 | 4.37 | 4.46 |
| | DBGNN (4,4) | 4.28 | 4.47 | 4.56 | 4.28 |
| workplace | DeepWalk | 2.23 | 1.85 | 1.48 | 2.23 |
| | Node2Vec p=1 q=4 | 3.3 | 3.11 | 2.95 | 3.3 |
| | GCN (32,16) | 8.67 | 8.6 | 9.61 | 8.26 |
| | EVO p=1 q=4 | 3.12 | 2.36 | 1.65 | 3.12 |
| | HONEM | 6.27 | 5.17 | 4.34 | 6.27 |
| | LGNN | 11.31 | 12.90 | 14.92 | 11.31 |
| | DBGNN (32,8) | 9.67 | 9.76 | 10.26 | 9.65 |

**Table 3:** Standard deviations of node classification in six dynamic graphs for static graph learning techniques (DeepWalk, node2vec, GCN) and time-aware methods (HONEM, EVO) as well as the DBGNN architecture proposed in this work.

## G   Ablation study of DBGNN without first-order message passing

As mentioned in the main text, our choice of performing message passing on the first-order graph in addition to the message passing in the higher-order De Bruijn graph is based on the idea that we want to additionally include information on the graph topology of the underlying system. To justify this design choice, we performed an ablation study, in which we removed the first-order message passing layer (see Figure 2) and evaluate the node classification performance of the resulting architecture, which we denote as DBGNN$^*$.

The results in Table 4 confirm that the inclusion of the first-order message passing layer considerably improves the performance of the DBGNN architecture. That we do not see such an improvement in the synthetic **temp-clusters** data set, where the performance of DBGNN$^*$ and DBGNN is identical, can be understood based on the fact that in this case the static topology is a random graph (cf.

Appendix C), i.e. the first-order topology does not add any additional information. Moreover, the synthetic dynamic graph has been created such that the clusters can be perfectly predicted based on only a second-order DeBruijn graph.

| dataset | method | Balanced Accuracy | F1-score-macro | Precision-macro | Recall-macro |
|---|---|---|---|---|---|
| temp-clusters | DBGNN* (16,16) | 100.0 | 100.0 | 100.0 | 100.0 |
| | DBGNN (16,16) | 100.0 | 100.0 | 100.0 | 100.0 |
| high-school-2011 | DBGNN* (32,8) | 60.3 | 59.4 | 61.1 | 60.3 |
| | DBGNN (32,8) | 64.4 | 63.7 | 65.14 | 64.4 |
| high-school-2012 | DBGNN* (4,8) | 65.3 | 64.96 | 66.63 | 65.3 |
| | DBGNN (4,8) | 65.8 | 65.89 | 67.27 | 65.8 |
| hospital | DBGNN* (32,16) | 53.3 | 51.78 | 56.72 | 53.23 |
| | DBGNN (32,16) | 59.04 | 55.26 | 58.71 | 57.71 |
| student-sms | DBGNN* (4,4) | 59.41 | 59.47 | 61.56 | 59.41 |
| | DBGNN (4,4) | 60.6 | 60.89 | 62.55 | 60.6 |
| workplace | DBGNN* (32,8) | 76.07 | 73.18 | 74.08 | 74.17 |
| | DBGNN (32,8) | 83.13 | 81.06 | 81.52 | 81.75 |

**Table 4:** Results node classification performance in an ablation study, where we use a model DBGNN* that is identical to DBGNN, except for the missing first-order message passing layer.

# H   Comments on computational complexity

Due to their resemblance to higher-order Markov chains, the computational complexity of our higher-order De Bruijn graph architecture could be a possible concern regarding its practical applicability. To address those potential concerns, we include an empirical investigation of the number of edges in higher-order De Bruijn graphs for all data sets, which determines the computational complexity of the message passing scheme.

For Markov chain models of unconstrained sequences of $n$ nodes, the number of transitions (i.e. edges) in a $k$-th order model grows exponentially as $n^{k+1}$ (see blue lines in left column of Figure 5). The size of such models quickly becomes impractical for real data sets.

However, compared to higher-order Markov chain models for unconstrained node sequences, in our work we are concerned with higher-order De Bruijn graph models of causal walks in a given graph. Since most real-world graphs are sparse, the number of walks of length $k$ that are theoretically possible is severely reduced compared to the state space of a $k$-th order Markov chain. For a directed graph with $n$ nodes and binary adjacency matrix $\mathbf{A}$ the number of walks of exactly length $k$ is given as $\sum_{ij} A_{ij}^k \leq n^k$, where $\mathbf{A}^k$ is the $k$-th power of $\mathbf{A}$ (see orange lines in the right column of Figure 5). This size is the theoretical upper limit for the size of a $k$-th order De Bruijn graph model of causal paths in dynamic graphs that holds iff all possible walks in the underlying graph are realized in terms of a corresponding causal walk. In real data sets on dynamic graphs, we find that only a small fraction of possible walks is actually realized as causal walks, which further reduces the size of the higher-order De Bruijn graphs that are used for message passing (see blue line in right column of Figure 5).

In summary, we find that the number of edges in a higher-order De Bruin graph model grows slowly (or even decreases) as the order increases, which considerably reduces the computational complexity of our method, making it practically applicable in a wide range of systems. We further note that the complexity of the message passing is independent of the length of the time series on the dynamic graphs, which is merely used to calculate the statistics of causal walks of length $k$ that are represented in the $k$th order De Bruijn graph models.

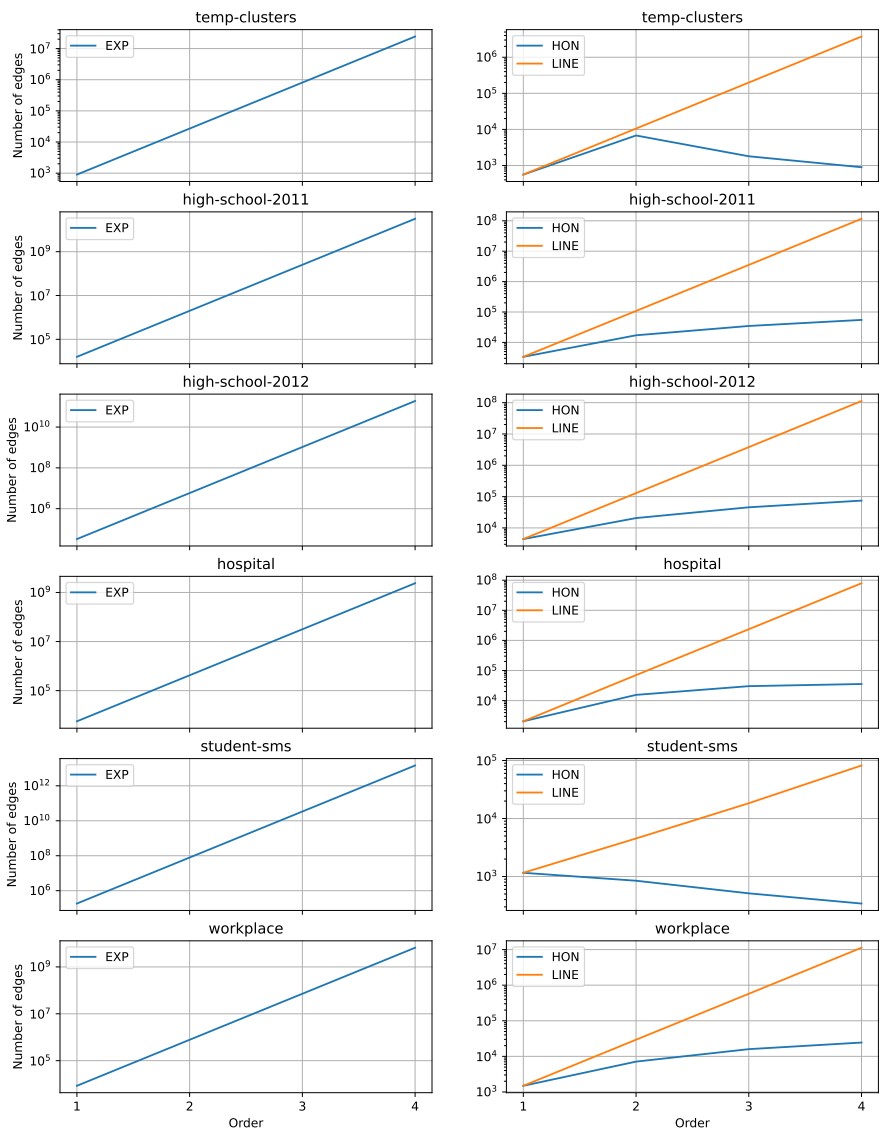

**Figure 5:** Number of edges in $k$-th order models for the six different data sets (rows). In the first column we show the number of edges in a $k$-th order Markov chain model for unconstrained sequences of $n$ nodes, which is given as $n^{k+1}$. The second column contains the number of edges in a repeated line graph construction of the first-order graph (orange line), and the number of edges in the actual $k$-th order De Bruijn graph model of causal walks observed in the data sets (blue line in second column). Due to the sparsity of the graphs and the relatively small number of long causal walks, we find that the actual size of the higher-order De Bruijn graph models are order of magnitudes smaller than the theoretical limit.

