# OpenReview forum: "De Bruijn goes Neural: Causality-Aware Graph Neural Networks for Time Series Data on Dynamic Graphs"
_logconference.io/LOG/2022/Conference — LoG 2022 Poster_

### Official Review · Reviewer_GkLL · 2022-10-15

**Overall Score:** 6
**Confidence:** 4

**Review:**

Summary:
   This paper proposes a causality aware GNN model for time-series data on dynamic graphs. It combines a statistical approach for causal topology discovery, a gradient-based learning for message passing, and a bipartite layer for learnt hidden representation mapping. The node classification results demonstrate its effectiveness on time series data.

Strengths:
This paper may be inspiring for augmented message passing scheme design on complicated graph structures.
Experiments on both empirical and synthetically generated dynamic graphs show that the explicit regularization of the message passing layer can considerably improve performance on node classification task.
The authors give clear definition about the causal walks and paths in dynamic graphs, which are the key concept of the proposed approach.
Weaknesses:
Lack of related work: the authors didn't give much background knowledge about what is De Bruign Graph.
The performance gain on the synthetic dataset seems much larger than the five empirical time series. The author didn't explain much on the reasons for this observation.
Concerns on scalability: the tested datasets are either relatively small on |V| or |E|. It is suggested that the author can include more discussion of the scalability of the proposed architecture.

---

### Official Review · Reviewer_akLc · 2022-10-21

**Overall Score:** 6
**Confidence:** 3

**Review:**

### Summary

The paper proposes a method (DBGNN) for learning from temporal information in dynamic networks where connections (edges) are associated with a timestamp, a problem commonly studied in network analysis but so far rarely treated in graph deep learning. These timestamps induce a causal structure on the graph, which the paper models in terms of a higher-order de Brujin graph. To leverage this causal structure in node classification, separate GNNs are run on the higher-order and original graphs, and the higher-order features are then distributed to the lower-order graph via a so-called "bipartite layer." The architecture is benchmarked on node classification in several real-world social interaction networks.

### Pros

* The paper appears to be the first to treat timestamped dynamic networks in a GNN framework. The idea and problem statement are relatively novel to this community, and seems like a powerful enough idea to have broad downstream applications.

* The de Brujin graph construction seems quite elegant and is a clean and straightforward idea for integrating temporal information.

* The synthetic part of the experiments does a good job of highlighting the key ways in which the task differs from ML on static graphs.

### Cons

* The paper focuses on social interaction networks, and the more general applicability is unclear. I would have liked to see more concrete discussion or demonstration on diverse tasks.

* The experimental evaluation is weak. Although there are four experimental datasets, they are all of the same task. The analysis of the learned embeddings is extremely short and does not lead to any insights.

* Given the relative simplicity of the network architecture, the design choices seem insufficently explored and/or justified. For example. the bipartite layer could assign based on the first rather than last node; or perhaps use some combination of all assignments.

* Once the background has been established, the novelty and technical insight of the paper are less clear. The main contribution amounts to message-passing over an augmented graph which is (seemingly) often-used and well-understood in the network analysis literature.

### Presentation

* The terms "causal topology," "causality," etc are misleadingly used since the focus is on temporal correlation and no causality is assumed or inferred. In any case, these terms are not properly defined in the paper.

* Please clearly state (ideally in the intro) the precise notion of dynamic network, i.e., edges exist only at point or interval of time. Currently the notion is left vague in the intro and makes the motivation of the paper hard to grasp until much later.

* Please discuss the baseline models EVO and HONEM in more technical detail, provide more details on the procedure for determining the correct order of the de Brujin graph, and define the "macro" metrics used in experiments.

* The term "non-Markovian" is often inappropriately used. It makes sense to describe the walks as non-Markovian, but not for message passing.

* Corrections: $\exists t \in \mathbb{N}: (v, w, t) \in E^\mathcal{T}$ (90), $\exists t_0, \ldots t_{l-1}$ (def 1), $\vec h_{u}^{k, l-1}$ (eq 1 RHS)

* 299-330 is too long for a single paragraph.

### Recommendation

I tentatively recommend acceptance. The paper brings to the attention of the community a family of methods and problems that could conceivably become an active area of study, which in my opinion outweights the weaknesses. However, once the background has been established, the novelty and technical insight of the paper are less clear, while the experiments are modest. My recommendation is conditional on the presentation and clarity issues being addressed.

---

### Official Review · Reviewer_35Vj · 2022-10-22

**Overall Score:** 6
**Confidence:** 5

**Review:**

This paper investigates the problem of modeling time series data on dynamic graphs. In this setting, a node A may only be able to influence aother node B if there is a "causal walk" from A to B, i.e., a temporally ordered sequence of edges from A to B. With the fact that a k-th order De Bruijin graph captures the causal graphs of length k-1 and k, this paper proposes a De Bruijin Graph Neural Network (DBGNN) model to augment the representations of each node with the causal graphs points to these nodes. The proposed method is shown to have superior empirical performance on serveral node classification tasks on dynamic graphs.

---

Strength:
1) This paper investigates an interesting problem of node classification on dynamic graphs. And the "causal walks" of node influence is indeed an important uniqueness of dynamic graphs that is worth modeling.
2) Leveraging De Bruijin graphs into GNNs seems to be novel.
3) The proposed method empirically performs well.
4) I very much like the figures 1 and 2.

---

Weakness and questions:

1) While I generally agree that modeling the "causal paths" can be helpful for prediction tasks on dynamic graphs, I feel that some of the design choices of the proposed DBGNN method are not well-motivated or empirically evaluated.

	1.1) What is the motivation of learning a representation for each "causal walk"? A simple alternative of using these paths I can think of is to directly guide the message-passing aggregations of node representations. This alternative seems to better reflect the idea of how one node can influence another node.
	1.2) What is the motivation of combining the first-order graph and the k-th order graph, since the first-order graph is known to be flawed? It would be nice to see an ablation study where one only uses the k-th order representation for the prediction task.

2) The computational complexity seems to be a big weakness for the proposed method, as the size of the De Bruijin graph grows quickly with k (although it may eventually shrink when the causal walk is too long).

3) The presentation could be improved. There are too much text on the background while the proposed method does not appear until page 5. I suggest the authors to move some of the background information to Appendix or near the end of the paper, as they are not the contribution and focus of this work. For example, I feel that the two paragraphs of "Non-Markovian characteristics of dynamic graphs" could be safely moved to Appendix without undermining the introduction of the proposed methodology.
4) A critical step of the proposed method, selecting the optimal k for which De Bruijin graph to use, is barely described. The paper only states that one can use the method in [12] to do the selection. Adding at least high-level intuition of the method in [12] will be appreciated for the completeness of this work.
5) Also relevant to the selection of k. The paper claims the statistical model selection method can address "the issue that De Bruijin graphs with different orders k can be used to model the same data set". However, according to the experiment section, this method seems to only be able to select from k=1 vs k=2?
6) How sensitive is the proposed method with respect to the hyperparameter delta and how to select it in practice?
7) Could the author also compare with Line Graph Neural Network (LGNN, https://docs.dgl.ai/en/0.9.x/tutorials/models/1_gnn/6_line_graph.html)? When k=2, the nodes in De Bruijin graph are just edges. It would be interesting to see how it compares to LGNN.
8) Finally, all the prediction targets in the empirical datasets are static properties of the nodes (e.g. gender or department). Could the authors elaborate on how these properties might be relevant to the "causal paths" of node influences, such that modeling these "causal paths" could be helpful to the prediction?


Minor question: line 267: "... the (possible) causal walks ...". What does "possible" refers to?

Typos:
- line 36: "..., e.g. for reachability ..." remove "for"
- line 128: "..., or embedding[9, ...]" missing space after embedding
- line 237: "... calculate the statistic ..." statistics

---

In general, this paper introduces an interesting idea of leveraging De Bruijin graph to model the "causal walks" in dynamic graphs. However, there is still a large room to improve the implementation and presentation of the idea.

---

### Meta-Review · Area_Chair_ftNy · 2022-11-20

**Confidence:** 4
**Recommendation:** Accept

**Meta Review:**

This paper proposes to turn dynamic graphs into De Bruijn graphs, where nodes are walks with length k-1 and edges are walks with length k. The idea is interesting and novel. The results are empirically good. There are concerns on scalability of the proposed approach. Even for sparse graph, when k is big, the potential nodes in the k-order graph will be overwhelming. Another concern is on the terminology. Causality is a well-defined research area, following the frameworks of Pearl or Rubin. In addition, the authors have made significant efforts in address the reviewers’ concerns, and the paper has been updated. I will weigh novelty before the limitations and recommend accept.

---

### Decision · Program_Chairs · 2022-11-22

Accept (Poster)